# Risk Factor Analysis for Proximal Junctional Kyphosis in Neuromuscular Scoliosis: A Single-Center Study

**DOI:** 10.3390/jcm14113646

**Published:** 2025-05-22

**Authors:** Tobias Lange, Kathrin Boeckenfoerde, Georg Gosheger, Sebastian Bockholt, Albert Schulze Bövingloh

**Affiliations:** 1Department of Orthopedics and Trauma Surgery, St. Josef-Hospital, Ruhr-University Bochum, 44791 Bochum, Germany; 2Department of Orthodontics, Heidelberg University Hospital, 69120 Heidelberg, Germany; kathrin.boeckenfoerde@med.uni-heidelberg.de; 3Department of Orthopedics and Tumor Orthopedics, Muenster University Hospital, 48149 Muenster, Germany; georg.gosheger@ukmuenster.de (G.G.); sebastian.bockholt@ukmuenster.de (S.B.); albert.schulzeboevingloh@ukmuenster.de (A.S.B.)

**Keywords:** neuromuscular scoliosis, neuromyopathic scoliosis, proximal junctional kyphosis, complication, risk factor, spinous process, sagittal alignment, rod contouring, hyperkyphosis

## Abstract

**Background/Objectives:** Proximal junctional kyphosis (PJK) is one of the most frequently discussed complications following corrective surgery in patients with neuromuscular scoliosis (NMS). Despite its clinical relevance, the etiology of PJK remains incompletely understood and appears to be multifactorial. Biomechanical and limited clinical studies suggest that preoperative hyperkyphosis, resection of the spinous processes with consequent disruption of posterior ligamentous structures, and rod contouring parameters may contribute as risk factors. **Methods**: To validate these findings, we retrospectively analyzed 99 NMS patients who underwent posterior spinal fusion using a standardized screw-rod system between 2009 and 2017. Radiographic assessments were conducted at three time points: preoperatively (preOP), postoperatively (postOP), and at a mean follow-up (FU) of 29 months. Clinical variables collected included patient age, weight, height, sex, and Risser sign. Radiographic evaluations encompassed Cobb angles, thoracic kyphosis (TK), lumbar lordosis, the levels of the upper (UIV) and lower (LIV) instrumented vertebrae, the total number of fused segments, parameters of sagittal alignment, the rod contour angle (RCA), and the postoperative mismatch between RCA and the proximal junctional angle (PJA). Based on the development of proximal junctional kyphosis, patients were categorized into PJK and non-PJK groups. **Results**: The overall incidence of PJK was 23.2%. In line with previous biomechanical findings, spinous process resection was significantly associated with PJK development. Furthermore, the PJK group demonstrated significantly higher preoperative TK (59.3° ± 29.04° vs. 34.5° ± 26.76°, *p* < 0.001), greater RCA (10.2° ± 4.01° vs. 7.7° ± 4.34°, *p* = 0.021), and a larger postoperative mismatch between PJA and RCA (PJA−RCA: 3.8° ± 6.76° vs. −1.8° ± 6.55°, *p* < 0.001) compared to the non-PJK group. **Conclusions**: Spinous process resection, a pronounced mismatch between postoperative PJA and RCA (odds ratio [OR] = 1.19, *p* = 0.002), excessive rod bending (i.e., high RCA), and severe preoperative thoracic hyperkyphosis with an expected increase in the risk of PJK of approximately 6.5% per degree of increase in preoperative TK are significant risk factors for PJK. These variables should be carefully considered during the surgical planning and execution of deformity correction in NMS patients.

## 1. Introduction

Neuromuscular scoliosis (NMS) is a complex three-dimensional spinal deformity caused by neuromyopathic disorders that impair muscle strength and control [1,2]. It is commonly associated with conditions such as spinal muscular atrophy, cerebral palsy, and muscular dystrophy, leading to progressive spinal misalignment and instability [3]. Treatment strategies for NMS include conservative approaches and surgical interventions, such as posterior spinal correction and fusion [4], as well as the use of growth-friendly devices like VEPTR™ or MAGEC™ rods in younger patients [5,6,7,8].

Surgical correction plays a central role in halting progression, preventing complications, and improving quality of life in patients with NMS [1,9,10]. However, these procedures are associated with a high rate of complications, necessitating close postoperative monitoring [11,12,13]. A meta-analysis by Sharma et al. (2012) demonstrated significantly higher complication rates in patients with NMS compared to those with adolescent idiopathic scoliosis (AIS) [14,15,16].

One notable postoperative complication is proximal junctional kyphosis (PJK), which can result in pain, spinal imbalance, implant-related problems, and even neurological deficits [17,18]. Reported incidence rates for PJK vary widely between 7% and 61%, likely due in part to differing definitions used across studies [9,19]. While numerous investigations have examined PJK risk factors in adult deformity and AIS populations, dedicated studies on NMS remain limited [13,20,21].

In NMS patients, particularly children undergoing long spinal fusions to the sacrum and pelvis, PJK is recognized as a frequent and serious complication [22,23,24]. The spectrum of the condition ranges from mild PJK to proximal junctional failure (PJF), underlining the importance of a comprehensive understanding of its causes and prevention [23,25]. Contributing factors discussed in the literature include surgical techniques, choice of instrumentation (e.g., pedicle screws vs. hooks or sublaminar bands), and rod material stiffness. In addition, sagittal profile parameters, particularly preoperative thoracic hyperkyphosis, have been linked to PJK development [26,27,28,29,30,31,32].

More recently, specific radiological parameters related to rod contouring have gained attention. In particular, the rod contour angle (RCA) and its mismatch with the proximal junctional angle (PJA) have been proposed as influential factors [28,33,34,35,36,37]. A pronounced curvature at the proximal rod end (i.e., high RCA) and a significant postoperative discrepancy between PJA and RCA (PJA–RCA) are suspected to contribute to junctional instability. However, these relationships remain poorly understood and are scarcely studied in the NMS population [33].

Furthermore, resection of spinous processes—commonly performed during surgery to harvest bone for fusion—inevitably disrupts the supra- and interspinous ligaments. This destabilization may increase flexibility in the adjacent cranial segments and has been identified in recent biomechanical studies as a potential risk factor for PJK [38].

Given the surgical complexity of NMS, especially in pediatric patients, the identification and understanding of such risk factors are critical. Incorporating them into preoperative planning and intraoperative decision-making may help to reduce the incidence of PJK and improve long-term outcomes [24].

Proximal junctional kyphosis remains a significant challenge in the surgical treatment of neuromuscular scoliosis. This single-center study aims to identify modifiable risk factors—such as spinous process resection and inadequate rod contouring—to support the development of preventive strategies. Ultimately, addressing these factors may contribute to reducing PJK incidence and enhancing surgical outcomes in this vulnerable patient population.

## 2. Materials and Methods

This retrospective single-center study analyzed a cohort of 99 patients diagnosed with neuromuscular scoliosis (NMS) who underwent corrective posterior spinal fusion between 2009 and 2017 using a standardized screw-rod system (titanium polyaxial pedicle screws, titanium and/or cobalt-chromium rods; Expedium^®^, DePuy Synthes, 325 Paramount Drive, Raynham, MA 02767, USA) and a uniform surgical technique. In 12 patients, bilateral transverse process hooks were utilized at the upper instrumented vertebra (UIV). Spinal bands were used in 10 patients: in 7 cases, three bands were applied on the concave side at the apex of the thoracic curve to assist curve correction in the presence of pedicle hypoplasia; in 3 cases, spinal bands were placed at the UIV. No additional prophylactic strategies—such as vertebroplasty or sublaminar wiring—were employed to prevent proximal junctional kyphosis (PJK). During the postoperative course, the patients did not receive any standardized rehabilitation measures.

Given the retrospective nature of the study, which was based on existing radiographic (hospital PACS) and clinical (patient chart) data, patient consent was not required in accordance with institutional regulations.

Full-spine standing or sitting radiographs (anteroposterior and lateral views) were obtained preoperatively (preOP), immediately postoperatively (postOP), and at a mean follow-up (FU) of 29 months. Some of the patients were able to sit unaided, some had to be held in a sitting position by relatives for X-ray imaging, and some were unable to sit even with assistance, so in this case, as an exception, X-ray imaging was performed while lying down. All images were digitally stitched. Radiographic measurements were performed by a single observer (K.B.), while clinical data were extracted from patient records.

Collected clinical data included age at surgery, sex, weight, height, and Risser sign. Radiographic assessments comprised preOP and postOP Cobb angles, thoracic kyphosis (T4–T12), lumbar lordosis (L1–S1), UIV and lower instrumented vertebra (LIV) levels, number of instrumented vertebrae, rod contour angle (RCA, defined as the angle between UIV and UIV–1), and mismatch between RCA and the proximal junctional angle (PJA, measured as the angle between the inferior endplate of UIV and the superior endplate of UIV+2). Additional sagittal spinopelvic parameters included pelvic tilt (PT), sacral slope (SS), pelvic incidence (PI), and sagittal vertical axis (SVA).

PJK was defined according to established criteria [39,40], requiring both of the following: (1) a segmental kyphosis of ≥10° between UIV and UIV+2, and (2) a postoperative increase in this segmental kyphosis of at least 10° compared to the preoperative value (Figure 1).

The study cohort was divided into two groups: patients with PJK (*n* = 23) and those without PJK (*n* = 76).

Statistical analysis was performed using SPSS Statistics, version 28.0.1.1 (IBM Corp., Armonk, NY, USA). The incidence of PJK was calculated using cross-tabulations. Group comparisons were conducted using the Student’s *t*-test or Mann–Whitney *U*-test for continuous variables, and the chi-square test for categorical variables. Pearson correlation analysis was used to explore relationships between significant risk factors. Variables with a statistically significant association with PJK were further analyzed using multivariate logistic regression. In cases of high intercorrelation between variables (|r| > 0.7), only the parameter with the stronger correlation to the dependent variable (PJK) was included in the final model. Odds ratios (OR) were calculated to quantify the influence of individual predictors on the likelihood of developing PJK. Descriptive data are reported as mean ± standard deviation (SD). A *p*-value of less than 0.05 was considered statistically significant.

The study received approval from the Ethics Committee of the Westfalen-Lippe Medical Association on 23 August 2017 (No: 2017-427-f-S) and complies with the guidelines of the Declaration of Helsinki.

## 3. Results

A total of 99 patients with neuromuscular scoliosis (NMS) were included in this study. The underlying diagnoses were as follows: cerebral palsy (*n* = 54), muscular dystrophy (*n* = 17), spinal muscular atrophy (*n* = 13), spina bifida (*n* = 10), neurofibromatosis (*n* = 4), and other etiologies (*n* = 1). The mean age at surgery was 17.0 ± 7.46 years, and 51.5% (*n* = 51) of patients were female.

Proximal junctional kyphosis (PJK) developed in 23 patients (23.2%), while 76 patients (76.8%) did not develop PJK during the follow-up period. The mean follow-up duration was 29 months. Among the PJK group, 78% (*n* = 18) developed PJK within the first 12 months, whereas 22% (*n* = 5) were diagnosed after more than 24 months.

The distribution of the upper instrumented vertebra (UIV) was as follows: T2 (22.2%), T3 (29.3%), T4 (22.2%), T5 (7.1%), T6 (6.1%), and T7 (2.0%). The lowest instrumented vertebra (LIV) was distributed as follows: T12 (2.0%), L1 (5.1%), L2 (4.0%), L3 (9.1%), L4 (19.2%), L5 (32.3%), and S2/Ilium (18.2%) (Figure 2). On average, 13.8 vertebrae and 12.8 segments were instrumented per patient.

Demographic, radiographic, and surgical parameters were compared between groups (Table 1).

No significant differences were found for age, sex, BMI, etiology of neuromuscular disease, number of instrumented vertebrae, or pre- and postoperative Cobb angles. However, significant differences were observed in thoracic kyphosis (TK), the rate of spinous process resection, rod contour angle (RCA), and mismatch between RCA and proximal junctional angle (PJA).

While 47.5% (*n* = 47) of patients received bilateral titanium rods, 24.2% (*n* = 24) cobalt–chromium rods, and 15.2% (*n* = 15) a hybrid construct, rod material was not significantly associated with PJK incidence. Similarly, the use of transverse process hooks (*n* = 12) or spinal bands at the UIV (*n* = 3) did not affect the frequency of PJK.

Spinous process resection showed a significant association with PJK: 87.0% of patients in the PJK group underwent this procedure, compared to 59.2% in the non-PJK group (*p* = 0.036).

Preoperative thoracic kyphosis was significantly higher in the PJK group (59.3° ± 29.04° vs. 34.5° ± 26.76°, *p* < 0.001) and remained elevated postoperatively and at 12-month follow-up. Moreover, the reduction in TK from preOP to postOP was more pronounced in the PJK group (−21.3° ± 19.28° vs. −10.7° ± 20.2°, *p* = 0.032).

Immediately postoperatively, both PJA and RCA differed significantly between groups (PJA: 14.0° ± 6.31° vs. 5.9° ± 6.49°, *p* < 0.001; RCA: 10.2° ± 4.01° vs. 7.7° ± 4.34°, *p* = 0.021). Notably, the mismatch between PJA and RCA (postOP PJA–RCA) was significantly higher in the PJK group (3.78° ± 6.75° vs. −1.8° ± 6.55°, *p* < 0.001), indicating a potential biomechanical risk factor for PJK.

To assess potential predictors of PJK, Pearson correlation analysis was conducted (Table 2). No predictors showed multicollinearity (r > |0.7|), and thus all were included in the multivariate logistic regression model.

Spinous process resection, preoperative thoracic kyphosis (TK), and the postoperative mismatch between PJA and RCA were significant predictors in the regression model (Table 3). Spinous process resection increased the likelihood of developing PJK by 6.3-fold (OR = 6.35, *p* = 0.036). A one-degree increase in PJA–RCA mismatch raised the PJK risk by 19% (OR = 1.19, *p* = 0.002), while each degree increase in preoperative TK raised the risk by 6.5% (OR = 1.07, *p* = 0.017) (Figure 3).

## 4. Discussion

Proximal junctional kyphosis (PJK), particularly its more severe manifestation—proximal junctional failure (PJF)—is a well-documented complication following corrective spinal surgery with posterior instrumentation in patients with neuromuscular scoliosis (NMS). PJK carries the potential risk of necessitating revision surgery and may significantly impact patients’ overall health and quality of life. Reported incidence rates of proximal junctional kyphosis (PJK) vary considerably between 7% and 61%, depending on the definitions applied in previous studies [9,19]. These discrepancies are largely attributable to differences in study populations, which include varying scoliosis types such as NMS, adolescent idiopathic scoliosis (AIS), and adult spinal deformity.

In contrast to adult and AIS populations, where numerous risk factors have been well established, fewer studies have focused on PJK in NMS, resulting in limited data. Furthermore, a lack of consensus regarding the definition of PJK complicates the interpretation of findings. For example, Sacramenta et al. assessed the reproducibility and interobserver reliability of the proximal junctional angle (PJA) measured at UIV/UIV+1 or UIV/UIV+2, concluding that using the first two vertebrae above the upper instrumented vertebra (UIV) yielded high reproducibility and agreement [41].

In our study, 23% of patients developed PJK, a rate consistent with findings reported in the literature. Nevertheless, the heterogeneity in definitions remains a significant challenge. Akosman et al. (2024) likened the state of PJK research to the Tower of Babel, reflecting the confusion caused by a lack of standardized terminology and criteria [42]. Future research must address both the precise definition and underlying mechanisms of PJK and PJF to enable meaningful comparisons across studies and to guide the development of preventive strategies.

Well-established risk factors for PJK include compromised posterior ligamentous structures, poor bone quality, and sagittal imbalance [26,43,44]. Additional studies implicate combined anterior–posterior instrumentation, halo gravity traction [21], aggressive flattening of TK [30], preoperative hyperkyphosis, and thoracoplasty as contributing factors [45,46,47,48].

Lange et al. [38] demonstrated in a biomechanical study that the rupture or dissection of the supra- and interspinous ligaments, particularly in the cranial segments adjacent to the instrumentation or even below, is associated with a higher incidence of PJK. This may be explained by the fact that these ligaments span multiple levels from one spinous process to another [38]. Similarly, a finite element study by Cammarata et al. revealed a positive correlation between ligament dissection and increased stress at the junctional area, ultimately leading to PJK [43,49].

Our data support these findings, showing a sixfold increase in PJK risk following resection of spinous processes. Such resections, often performed to harvest autograft or facilitate spinal release, should therefore be approached with caution—especially in patients deemed at high risk for PJK [21,22]. Several studies also suggest that augmentation of the posterior tension band using Mersilene tape or hybrid constructs (e.g., sublaminar polyester bands) at the proximal instrumentation level may reduce PJK risk [29,50,51].

In adult spinal deformity, stretching or rupture of posterior ligaments and spinous process fractures are described as potential failure mechanisms. However, unlike modifiable intraoperative factors, these risks are largely outside the surgeon’s control [23].

Rod contouring at the proximal end of the instrumentation is another frequently discussed risk factor. Parameters such as PJA and rod contour angle (RCA), as well as the mismatch between them, have been proposed as relevant indicators of risk. Intraoperative rod contouring directly affects adjacent segments [46]. Our findings confirm the biomechanical results of Cammarata et al., showing that increasing the RCA from 10° to 20°, 30°, and 40° raises the PJA by 6%, 13%, and 19%, respectively, highlighting that over-contouring of the rod increases the likelihood of PJK [43].

Additionally, our study reveals that each degree of RCA increase is associated with a 1.15-fold increase in PJK risk. Both excessive and insufficient contouring at the proximal end—particularly when not aligned with the patient’s native sagittal profile—appear problematic. Over-contouring results in an elevated PJA, which was significantly higher in the PJK group postoperatively. Conversely, a rod that is too straight also poses a risk, as demonstrated by a pronounced mismatch between PJA and RCA (i.e., RCA significantly lower than PJA). Our logistic regression analysis showed that each degree of mismatch increases the risk of PJK by 19%.

It is important to note that RCA is influenced by individual factors, including rod material, diameter, and screw angulation. Nonetheless, our results are consistent with those of Wang et al. [28], who identified inadequate rod contouring relative to thoracic kyphosis as a major risk factor. They emphasize the need for individualized and precise intraoperative rod shaping to mitigate PJK risk [28,52].

We initially investigated thoracic kyphosis (TK) to understand the impact of sagittal alignment on PJK. Preoperative TK was significantly greater in the PJK group compared to the non-PJK group, with the risk of developing PJK increasing by 6.5% for each additional degree of TK. These findings align with those reported by Kim et al. and Lonner et al. [45,46]. Consequently, surgeons should be cautious in cases of preoperative hyperkyphosis. Kim et al. [45] also found that aggressive TK correction during surgery significantly elevates PJK risk. While our data showed a similar trend, statistical significance was not achieved. However, the extent of TK correction differs significantly between PJK and non-PJK patients [45].

Regarding the sagittal vertical axis (SVA), no significant difference was found between groups in our cohort, consistent with the findings of Burton et al. [53]. In contrast, Wang et al. reported preoperative SVA differences between PJK and non-PJK groups [28]. In our study, baseline SVA values were similar, potentially due to the characteristic mobility impairments of NMS patients, which complicate standardized radiographic imaging. Differences in X-ray protocols across institutions may further contribute to conflicting findings [54,55].

This study has several limitations. The lack of standardized radiographic acquisition, combined with varying patient positioning due to physical impairments, introduces potential sources of error. While whole-body imaging systems like EOS^®^ could help address these challenges, such technologies are not universally available [56,57]. Additionally, the retrospective design introduces inherent biases. Ideally, radiographic assessments should be complemented by clinical data to evaluate the functional significance of PJK and potential PJF. Our study focuses exclusively on radiographic parameters and does not allow conclusions regarding clinical outcomes. A further limitation of the study is the incomplete data regarding preoperative mobility status, particularly the ability to sit freely, based on the records.

Rod contouring and its relationship to native kyphosis—quantified by RCA–PJA mismatch—may influence clinical outcomes, but further prospective studies are needed. Tools such as the Hart-ISSG severity scale may help correlate radiographic findings with clinical relevance [58]. Furthermore, the literature lacks consensus on which Cobb angle defines clinically significant PJK. Bridwell et al. proposed a 20° threshold in adults, yet found no corresponding difference in clinical outcome scores. Interestingly, they observed that patients with shorter instrumented segments ending in the lower thoracic spine, and those with iliac screws extending to the pelvis, had higher PJK angles, which may also constitute risk factors—at least in adult populations [59].

However, our study presents one of the largest single-center NMS cohorts analyzed to date. By including only complete and recent patient data, and avoiding inter-center variability in surgical technique, we provide a robust basis for future research focused specifically on NMS-related PJK.

## 5. Conclusions

The present study delineates several significant risk factors associated with the development of proximal junctional kyphosis (PJK) in patients diagnosed with neuromuscular scoliosis (NMS). These include intraoperative resection of the spinous processes, elevated preoperative thoracic kyphosis, excessive absolute rod contour angle (RCA), and a pronounced mismatch between the proximal junctional angle (PJA) and RCA. This mismatch reflects suboptimal rod contouring that does not align with the patient’s native thoracic sagittal profile.

However, it is important that future and prospective clinical studies verify the results of biomechanical and clinical retrospective studies.

To minimize the risk of PJK, surgeons should avoid destabilizing adjacent segments through aggressive resection of the spinous processes, particularly when harvesting autologous bone for spondylodesis. Moreover, careful attention must be given to rod contouring—especially at the proximal end of the instrumentation—to ensure it accurately reflects the patient’s individual sagittal alignment. This is particularly important in cases of preoperative thoracic hyperkyphosis, where precise adaptation to the natural high-thoracic profile, as indicated by the PJA, is essential for preventing junctional complications.

## Figures and Tables

**Figure 1 jcm-14-03646-f001:**
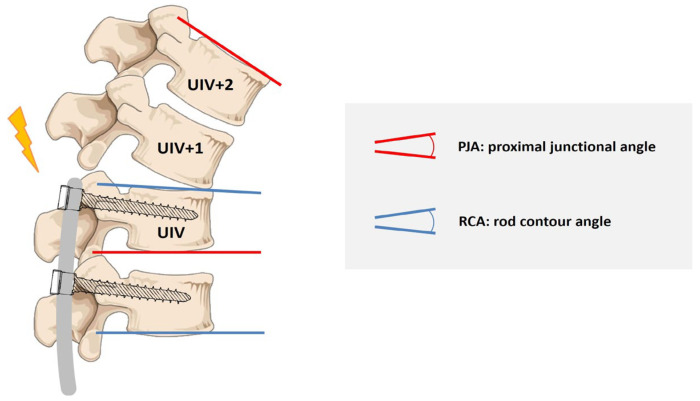
Measurement of proximal junctional kyphosis (PJK) and rod contour angle (RCA).

**Figure 2 jcm-14-03646-f002:**
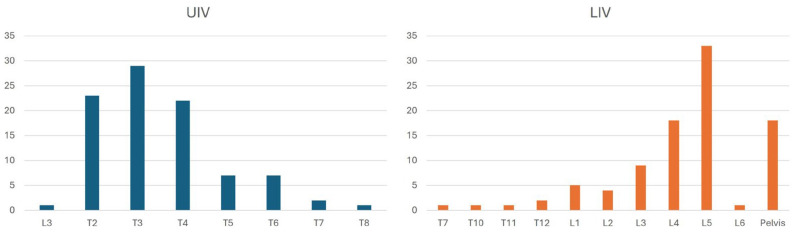
Distribution of upper (UIV) and lower (LIV) instrumented vertebrae.

**Figure 3 jcm-14-03646-f003:**
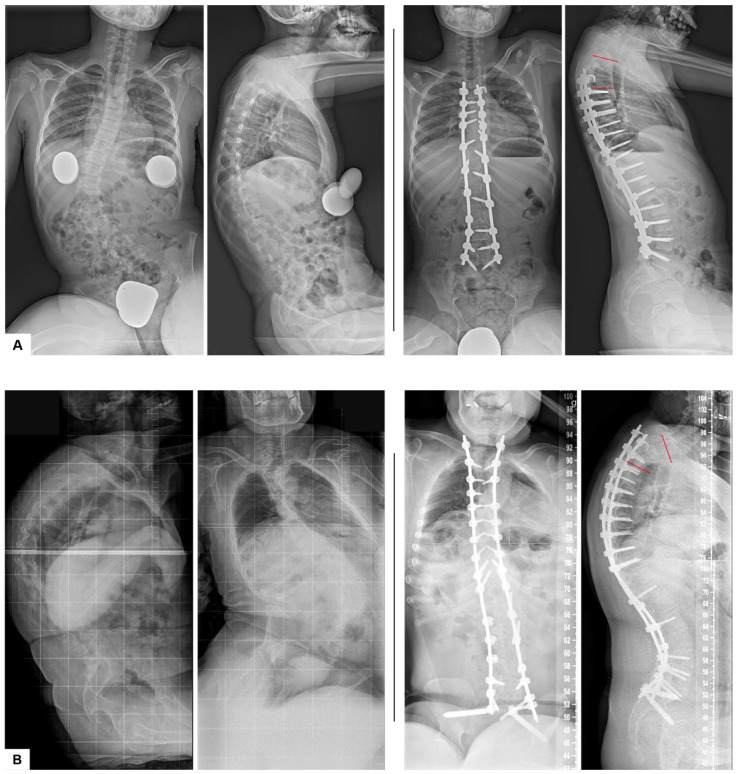
Representative radiographs from the non-PJK (**A**) and PJK (**B**) groups. PJA indicated by red lines.

**Table 1 jcm-14-03646-t001:** Demographic, radiographic, and surgical parameters. Values are mean ± standard deviation (SD) unless otherwise stated.

	Non-PJK Group	PJK Group	*p*
age	16.6 ± 6.78	18.5 ± 9.36	0.260
BMI	18.7 ± 5.18	17.4 ± 4.78	0.282
UIV (median)	T3	T3	0.308
LIV (median)	L5	L5	0.244
instrumented vertebra (*n*)	13.8 ± 2.72	14.0 ± 2.47	0.770
fused segments (*n*)	12.7 ± 2.71	13.5 ± 2.09	0.190
percentage of patients with resected spinous processes	59.2%	87.0%	**0.036**
resected spinous processes (*n*)	7.2 ± 5.65	9.4 ± 4.26	0.101
Cobb preOP (°)	81.7 ± 26.38	80.7 ± 26.41	0.874
Δ Cobb preOP vs. postOP (°)	−42.3 ± 17.84	−44.3 ± 17.6	0.645
TK preOP (°)	34.5 ± 26.76	59.3 ± 29.04	**<0.001**
TK postOP (°)	24.1 ± 14.11	38.0 ± 16.48	**<0.001**
TK 12 m FU (°)	24.4 ± 15.15	39.2 ± 18.51	**<0.001**
Δ TK preOP vs. postOP (°)	−10.7 ± 20.2	−21.3 ± 19.28	**0.032**
LL preOP (°)	37.9 ± 28.11	41.4 ± 31.91	0.625
LL postOP (°)	37.1 ± 16.59	44.0 ± 11.58	0.067
LL 12 m FU (°)	37.7 ± 18.64	42.6 ± 17.23	0.286
PI (°)	54.6 ± 22.3	48.4 ± 10.55	0.486
PT (°)	10.2 ± 8.76	7.3 ± 3.35	0.387
SL (°)	44.4 ± 20.51	41.1 ± 11.74	0.691
SVA preOP (cm)	0.4 ± 5.98	0.6 ± 2.85	0.919
SVA postOP (cm)	0.1 ± 3.82	1.2 ± 5.09	0.671
SVA 12 m FU (cm)	1.0 ± 6.35	−2.4 ± 2.85	0.139
PJA preOP (°)	3.7 ± 7.93	4.2 ± 7.22	0.777
PJA postOP (°)	5.9 ± 6.49	14.0 ± 6.31	**<0.001**
PJA 12 m FU (°)	7.3 ± 8.91	21.3 ± 10.02	**<0.001**
RCA (°)	7.7 ± 4.34	10.2 ± 4.01	**0.021**
postOP PJA-RCA (°)	−1.8 ± 6.55	3.78 ± 6.75	**<0.001**

Note. UIV, upper instrumented vertebra; LIV, lowest instrumented vertebra; TK, thoracic kyphosis (T4–12); LL, lumbar lordosis (L1–S1); PI, pelvic incidence; PT, pelvic tilt; SL, sacral slope; SVA, sagittal vertical axis; PJA, proximal junctional angle; RCA, rod contour angle; postOP PJA–RCA, mismatch between postOP PJA and RCA.

**Table 2 jcm-14-03646-t002:** Pearson correlation matrix among predictors.

	1.	2.	3.	4.	5.	6.
Spinous process resection	1					
2. TK (preOP)	0.044	1				
3. ΔTK (preOP vs. postOP)	0.088	0.862 **	1			
4. RCA (postOP)	0.001	0.338 **	−0.218 *	1		
5. ΔPJA-RCA (postOP)	0.142	0.005	0.014	−0.240 *	1	
6. PJK	0.224 *	0.377 **	−0.235 *	0.248 *	0.355 **	1

* *p* < 0.05, ** *p* < 0.01. Note. ΔPJA-RCA (postOP), mismatch between postOP PJA and rod contour angle; RCA, rod contour angle; TK, thoracic kyphosis T4–T12; ΔTK, thoracic kyphosis preOP–thoracic kyphosis postOP.

**Table 3 jcm-14-03646-t003:** Multivariate logistic regression analysis of risk factors for PJK.

	Estimate Coefficient	Odds Ratio (Exp(B))	*p*	95% Confidence Interval	
Spinous process resection	1.848	6.346	0.036	0.769	52.396
TK (preOP)	0.063	1.065	0.017	1.011	1.121
ΔTK (pre vs. postOP)	0.043	1.043	0.175	0.981	1.110
RCA (postOP)	0.137	1.147	0.094	0.977	1.346
ΔPJA-RCA (postOP)	0.174	1.190	0.002	1.065	1.331

Note. ΔPJA-RCA (postOP), mismatch between postOP PJA and rod contour angle; RCA, rod contour angle; TK, thoracic kyphosis T4–T12; ΔTK, thoracic kyphosis preOP–thoracic kyphosis postOP.

## Data Availability

Data collected for this study, including individual patient data, will not be made available.

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
