# Peer review of "Risk Factor Analysis for Proximal Junctional Kyphosis in Neuromuscular Scoliosis: A Single-Center Study"

_jcm, 2025, doi:10.3390/jcm14113646_

Round 1

Reviewer 1 Report

Comments and Suggestions for Authors

Dear Authors,

Thank you for submitting research work. I found the topic you address to be of significant importance within the field of spinal deformity, particularly concerning neuromuscular scoliosis. 

To further strengthen your manuscript, I have a few suggestions regarding the information presented. Specifically, it would be beneficial to include details about the patients' rehabilitation history. Could you please provide information on whether the patients underwent any rehabilitation, either pre-operatively or post-operatively? If so, specifying the frequency and type of rehabilitation interventions (mayby methods of therpay eg. Bobath/Vojta) would be highly valuable. I believe this information is pertinent as it could potentially influence sagittal alignment and the subsequent development of proximal junctional kyphosis (PJK) in this patients.

Furthermore, to provide a more comprehensive understanding of the patient cohort, I would appreciate clarification on the patients' mobility status. Could you please indicate whether the patients were wheelchair users capable of maintaining active sitting independently? This information could be included in the description of the study group in material and methods chapter or, if this data is unavailable, acknowledged as a limitation within the manuscript.

Incorporating these details would doubtless enhance the interpretability of your findings.

Sincerely,

Author Response

Comments 1: Specifically, it would be beneficial to include details about the patients' rehabilitation history. Could you please provide information on whether the patients underwent any rehabilitation, either pre-operatively or post-operatively? If so, specifying the frequency and type of rehabilitation interventions (mayby methods of therpay eg. Bobath/Vojta) would be highly valuable. I believe this information is pertinent as it could potentially influence sagittal alignment and the subsequent development of proximal junctional kyphosis (PJK) in this patients.

Response 1: Thank you for pointing this out. You are absolutely right that postoperative rehabilitation could potentially influence the development of PJK. However, in our clinic, we either did not provide any specific rehabilitation measures to the patients at all, or if rehabilitation was provided in individual cases, it was not standardized by us in any way. Therefore, unfortunately, we do not have any information on this. We have now mentioned this fact in the manuscript:

"During the postoperative course, the patients did not receive any standardized rehabilitation measures." [page 3, ll. 102-103]

Comments 2: Furthermore, to provide a more comprehensive understanding of the patient cohort, I would appreciate clarification on the patients' mobility status. Could you please indicate whether the patients were wheelchair users capable of maintaining active sitting independently? This information could be included in the description of the study group in material and methods chapter or, if this data is unavailable, acknowledged as a limitation within the manuscript.

Response 2: Dear reviewer, thank you for your comment. You are absolutely right; information on mobility and the ability to sit actively would have been extremely valuable. In reviewing the available data based on the records, we unfortunately have to conclude that we cannot reliably determine the preoperative mobility status of a significant proportion of patients.

However, we have included the following passage in the materials and methods section:

"Some of the patients were able to sit unaided, some had to be held in a sitting position by relatives for X-ray imaging, and some were unable to sit even with assistance, so in this case, as an exception, X-ray imaging was performed while lying down." [p. 3, ll. 109-112]

Furthermore, we have included this fact in the limitations as follows:

"A further limitation of the study is the incomplete data regarding preoperative mobility status, particularly the ability to sit freely, based on the records." [page 9, ll. 313-315]

Reviewer 2 Report

Comments and Suggestions for Authors

This manuscript attempts to summarize our existing knowledge regarding the entity of junctional kyphosis pathophysiology,risk factors and measurements that could be implemented in order to eliminate its overall incidence. This means that the main question addressed by the
article is to highlight the relevant risk factors and to mention the possible strategies that could reduce this complication. This topic is relevant to the field, although it cannot be considered as  original because several relevant studies have been published. Nevertheless, it attempts to address a specific gap in the field, as there is a lack of cumulative evidence based on studies that included a considerable number of patients.  Although its results are in concordance with other published material, this article  adds to the subject area the findings of a retrospective analysis which are based on a significant number of patients suffering from neuromuscular scoliosis. Since this article is based on the results of a single reference center, no specific improvements should be considered by the authors regarding the methodology. Hopefully,  further controls should be considered, including the acquisition of prospective data. However, the scientific soundness of this article cannot be overlooked. The conclusions seem to be consistent with the evidence and arguments presented and they address the main question that is posed, as they are consistent with the pathophysiology of this entity and the previously reported data. All the references are appropriate and, To the best of my knowledge, all relevant to the field references are included and no one of them is inappropriate. Nevertheless, I strongly believe that the title of the manuscript should be modified in order to underline the fact that this risk factor analysis is based on a single center data. Finally, any additional comments on the tables and figures did not seem to be required..

Author Response

Comments 1: Nevertheless, I strongly believe that the title of the manuscript should be modified in order to underline the fact that this risk factor analysis is based on a single center data. 

Response 1: We fully agree with this comment. Therefore, we have changed the title highlighting, that this analysis is based on a single center study: 

"Risk Factor Analysis for Proximal Junctional Kyphosis in Neuromuscular Scoliosis: A Single Center Study"

Reviewer 3 Report

Comments and Suggestions for Authors
  1. Including a list of abbreviations will improve readability for the reader.
  2. Highlighting the significant p-values (making it bold) in Table 1 will enhance readability.
  3. Maintain consistent verb tense throughout the manuscript. Use present tense when describing the paper’s content and past tense when referring to methods or actions taken.
  4. The term "severe preoperative thoracic hyperkyphosis" is used in conclusions but not defined (e.g., specific degree threshold). Clarifying what constitutes "severe" would improve precision.
  5. The introduction identifies a gap in NMS-specific PJK studies but does not explicitly state why prior biomechanical findings require clinical validation. A sentence explaining this would strengthen the study’s justification.
  6. The study includes 99 patients with diverse NMS etiologies. However, it does not report whether these subgroups were analyzed for differences in PJK incidence. Given potential variations in muscle tone or bone quality, subgroup analysis could strengthen the findings.
  7. The conclusion states that PJA-RCA mismatch is the “most critical” risk factor, but the ORs suggest spinous process resection has a stronger effect (OR=6.346 vs. OR=1.19). This claim should be revised to reflect the data.
  8. The conclusions do not propose specific future research directions (e.g., prospective studies, clinical outcome correlations). Including these would enhance the manuscript’s forward-looking impact.

Author Response

Comment 1: Including a list of abbreviations will improve readability for the reader.

Response 1: Dear reviewer, thank you very much for your comment.  We have included a list of abbreviations on page 10. 

Comment 2: Highlighting the significant p-values (making it bold) in Table 1 will enhance readability.

Response 2: Thank you for pointing this out. We have made the changes in table 1.

Comment 3: Maintain consistent verb tense throughout the manuscript. Use present tense when describing the paper’s content and past tense when referring to methods or actions taken.

Response 3:

Dear reviewer, we have reviewed the entire manuscript and have now corrected the verb tenses. We used the present tense when describing paper's content and past tense when referring to papers published in the past. Furthermore, past tense was used in the materials and method sections.

Comment 4: The term "severe preoperative thoracic hyperkyphosis" is used in conclusions but not defined (e.g., specific degree threshold). Clarifying what constitutes "severe" would improve precision.

Response 4:

Dear reviewer, thank you for your comment. There is no clear threshold in this case, but rather a relationship between thoracic hyperkyphosis and PJK, with an expected increase in the risk of PJK of approximately 6.5% per degree of increase in preoperative thoracic hyperkyphosis (see p. 6, lines 211-212).

However, we have further clarified the term "severe":

"...and severe preoperative thoracic hyperkyphosis with an expected increase in the risk of PJK of approximately 6.5% per degree of increase in preoperative TK are significant risk factors for PJK." (see p. 1, lines 36-37) 

Comment 5: The introduction identifies a gap in NMS-specific PJK studies but does not explicitly state why prior biomechanical findings require clinical validation. A sentence explaining this would strengthen the study’s justification.

Response 5: 

Dear reviewer, thank you for your comment. It's not entirely clear to us which part of the introduction you're referring to. In our opinion, we also highlighted on page 2 that there are several reasons to take a separate look at NMS patients, as previous literature has largely focused on AIS or adult scoliosis patients:
- NMS patients generally have more complications. Does this also apply to PJK?
- In addition to scoliosis, NMS patients often also have significant hyperkyphosis due to muscular impairment. Is this a risk factor for PJK?
- In principle, it is desirable to adapt surgical strategies in this patient group as well, if necessary, to reduce the risk of PJK, especially since these patients often also have an increased perioperative risk and therefore revision surgery should be avoided whenever possible—as is the case with all of our patients.

If you let us know specifically where you would like clarification or better explanation, we will be happy to implement this.

Comment 6: The study includes 99 patients with diverse NMS etiologies. However, it does not report whether these subgroups were analyzed for differences in PJK incidence. Given potential variations in muscle tone or bone quality, subgroup analysis could strengthen the findings.

Response 6:

Dear reviewer, thank you for your comment. We had already conducted this analysis, but didn't include it in the results. We have now included it and and believe this question is indeed very interesting and should be addressed in future prospective studies:

"No significant differences were found for age, sex, BMI, etiology of neuromuscular disease, number of instrumented vertebrae, or pre- and postoperative Cobb angles." (see p. 5, lines 174-175) 

Comment 7: The conclusion states that PJA-RCA mismatch is the “most critical” risk factor, but the ORs suggest spinous process resection has a stronger effect (OR=6.346 vs. OR=1.19). This claim should be revised to reflect the data.

Response 7:

Dear reviewer, thank you for your comment. You are absolutely right. The term "most critically" is misleading, and we have therefore deleted it.

Comment 8: The conclusions do not propose specific future research directions (e.g., prospective studies, clinical outcome correlations). Including these would enhance the manuscript’s forward-looking impact.

Response 8:

Dear reviewer, thank you for your comment. We added the following sentence to the conclusions section:

"However, it is important that future and prospective clinical studies verify the results of biomechanical and clinical retrospective studies." (see p. 9, lines 337-338)